# HGAMLP: A Scalable Training Framework for Heterogeneous Graph Learning

## Abstract

Heterogeneous graphs contain rich semantic information that can be exploited by heterogeneous graph neural networks (HGNNs). However, scaling HGNNs to large graphs is challenging due to the high computational cost. Existing scalable HGNNs use general subgraph construction method and mean aggregator to reduce the complexity. Despite their high scalability, they ignore two key characteristics of heterogeneous graphs, leading to low predictive performance. First, they adopt a fixed knowledge extractor during the local feature aggregation and the global semantic fusion of multiple meta-paths. Besides, they bury the graph structure information of the higher-order meta-paths and fail to fully leverage the higher-order global information. In this paper, we address these two limitations and propose a scalable HGNN framework called Heterogeneous Graph Attention Multi-Layer Perceptron (HGAMLP). Our framework employs a local multi-knowledge extractor to enhance the node representation, and leverages the de-redundancy mechanism to extract the pure graph structure information from higher-order meta-paths. Besides, it also adopts a node-adaptive weight adjustment mechanism to fuse the global knowledge from each local knowledge extractor. We evaluate our framework on five commonly used heterogeneous graph datasets and show that it outperforms the state-of-the-art baselines in both accuracy and speed. Notably, our framework achieves the best performance on the large public heterogeneous graph dataset (i.e., Ogbn-mag) of Open Graph Benchmark [1].

## 1 Introduction

Heterogeneous graphs have different types of nodes and edges and rich semantic information. They are ubiquitous in many domains, such as traffic network Hu et al. (2019); Li et al. (2019), biology Vretinaris et al. (2021); Do et al. (2019), and relational databases Cvitkovic (2020); Cai et al. (2021), etc. As powerful models for learning from heterogeneous graphs, heterogeneous graph neural networks (HGNNs) have aroused lots of concern in both academia and industry in recent years. Despite the success of HGNNs Wang et al. (2019); Hu et al. (2020); Lv et al. (2021); Schlichtkrull et al. (2018) on small or medium-scale graphs, scaling them to large graphs is challenging due to the high computational and storage costs of feature propagation and attention mechanisms.

Existing scalable GNNs for homogeneous graphs Wu et al. (2019); Zhu & Koniusz (2021); Chen et al. (2020); Zhang et al. (2021); Sun et al. (2020) are not suitable for heterogeneous graphs, as they lose important semantic information. To address this issue, recent works such as Nars Yu et al. (2020) and SeHGNN Yang et al. (2023) decouple feature aggregation from model training by using mean aggregator instead of attention. This way, they only need to perform feature aggregation once before training, which reduces the computational overhead significantly. However, these methods also suffer from two major limitations:

**Fixed knowledge extractor.** HGNNs generally use the attention mechanism to aggregate neighbor information, but this mechanism is not scalable to large graphs due to its high computational and memory costs. SeHGNN and Nars have proposed to use the fixed mean aggregator instead of the attention mechanism (HAN, HGB) in neighbor aggregation, which they call the *fixed local knowledge extractor*. However, this approach ignores the fact that diverse local knowledge extractors

---

[1]https://ogb.stanford.edu/docs/leader_nodeprop

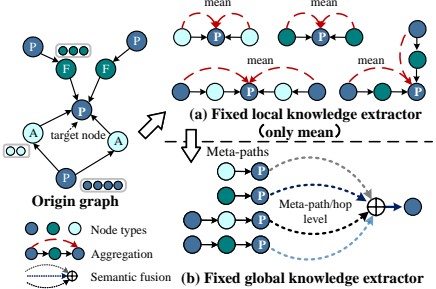

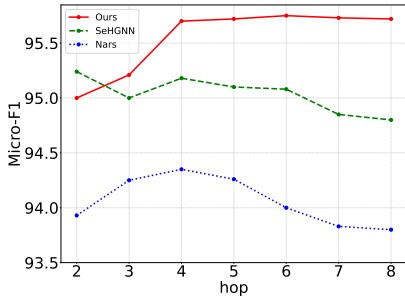

Figure 1: Fixed local and global knowledge extractor.

Figure 2: The effectiveness of de-redundancy mechanism.

can acquire varying graph structural information. Figure 1 (a) illustrates this problem. For each meta-path, existing methods only use mean aggregator without considering the personalized neighborhood feature aggregation. Moreover, most existing HGNNs Schlichtkrull et al. (2018); Zhang et al. (2019); Wang et al. (2019); Fu et al. (2020); Yu et al. (2020) fuse the semantic information in a coarse-grained manner. As shown in Figure 1 (b), they usually assign weights from the meta-path/hop level without considering the personalized combination of semantic information for each node. The above methods ignore the semantic fusion from a finer-grained node level, which we call the *fixed global knowledge extractor*. As the receptive field expands, each node receives more global information. However, it is hard for them to find the optimal combination of global information for each node.

**Buried global information.** Existing scalable HGNNs employ a general subgraph construction method that pre-defines the hop length and expands the receptive field by utilizing all proper meta-paths that are no longer than this length. However, this method has a limitation in capturing global information and further expanding the receptive field, ultimately failing to fully exploit the higher-order meta-paths. As Figure 2 shows, the accuracy of these methods either drops or plateaus as the number of hops increases. The reason is that the meta-paths generated by this method contain not only the current order information but also all the lower-order information related to that meta-path. Thus, existing scalable methods essentially amplify low-order graph structure information as they iteratively generate higher-order meta-paths. As the meta-path order increases, a significant amount of redundant low-order information is introduced, which buries the pure graph structure information of the higher-order meta-paths.

To address the above two limitations, we propose a new scalable HGNN framework, Heterogeneous Graph Attention Multi-Layer Perceptron (HGAMLP). HGAMLP is based on two observations: 1) *the high-order meta-path information is obscured by a large number of low-order sub-meta-paths*, and 2) *each node should have adaptive extractions of both local and global knowledge*. We analyze these two observations in depth in Section 3. Through these two observations, HGAMLP introduces a de-redundancy mechanism that masks all relevant low-order meta-path information and eliminates redundant graph structure information in high-order meta-paths. Besides, HGAMLP proposes local multi-knowledge extractors and a simple yet effective node-adaptive weight adjustment mechanism for local and global knowledge.

In summary, the core contributions of this paper are 1) **New Findings**. To the best of our knowledge, we are the first to explore the redundancy properties of high-order meta-paths and the limitations of fixed local/global knowledge extractors in existing scalable HGNNs. 2) **New Method**. We propose a scalable HGNN training framework named HGAMLP with a de-redundancy mechanism to extract the pure high-order graph structure information. Besides, we employ a local multi-knowledge extractor and a node-adaptive weight adjustment mechanism to fuse local and global knowledge. 3) **SOTA Performance**: HGAMLP outperforms state-of-the-art scalable HGNNs in both accuracy and training speed on four middle-scale datasets and one large-scale dataset.

## 2 PRELIMINARIES

In this section, we first introduce the notations and problem formulation, then review some existing works related to HGNNs and the scalability of GNNs.

## 2.1 Notations and Problem Formulation

In this paper, we consider a heterogeneous graph $G = (V, E, \phi, \psi)$, where $V$ is the set of nodes and $E$ is the set of edges. $\phi(v)$ denotes the node type of each node $v$, $\psi(e)$ denotes the edge type of each edge $e$. The set of all node types is denoted by $T_v = \{\phi(v), \forall v \in V\}$, the set of all edge types is denoted by $T_e = \{\psi(e), \forall e \in E\}$. Each node has a associated node type object $o_n = \{\phi(v_n) : v_n \in V\} \in T_v$. Each edge has a associated relation type $o_i, o_j = \{o_i \leftarrow o_j : o_i, o_j \in T_v\} \in T_e$, where $\leftarrow$ indicates the node type $o_j$ to the target node type $o_i$. A graph is considered as heterogeneous when $|\phi| + |\psi| > 2$.

Given a relation type $o_i \leftarrow o_j$ in $G$, the corresponding subgraph can be represented by the adjacency matrix $\mathbf{A}_{o_i, o_j} \in \mathbb{R}^{|V_{o_i}| \times |V_{o_j}|}$, where $|V_{o_i}|$ and $|V_{o_j}|$ denotes the number of nodes of node type $o_i$ and $o_j$. The index of each non-zero element in the adjacency matrix $\mathbf{A}_{o_i, o_j}$ implies there exists an edge. To capture more semantic information for the target node type, researchers can either manually specify the meta-path list $\Phi$ or repeatedly iterate over different relation types to generate the meta-path list $\Phi$. Specifically, each meta-path in $\Phi$ with $k$ hops between target node type and source node type is defined in the form $o_t \leftarrow \cdots \leftarrow o_{k-1} \leftarrow o_s$ (abbreviated as $o_t, ..., o_s$). For instance, an academic graph has the meta-path "$conference \leftarrow paper \leftarrow author$", which could indicate "the author writes the paper and published it in the conferenc". We follow the meta-path-based method to generate the node embedding and evaluate it for the node classification task.

## 2.2 Heterogeneous Graph Neural Networks

Heterogeneous Graph Neural Networks (HGNNs) can be broadly classified into two categories: meta-path-based method and meta-path-free method. The meta-path-based methods explicitly specify the path for message propagation, they use predefined paths that specify the type of relationships between nodes to guide the message-passing process between nodes, such as RGCN Schlichtkrull et al. (2018), HetGNN Zhang et al. (2019), HAN Wang et al. (2019), MAGNN Fu et al. (2020). On the other hand, the meta-path-free methods implicitly embed edge and node-type information into the propagated messages, they do not use predefined meta-paths for message propagation. Instead, they learn to combine the embeddings of node and edge types in the graph, such as RSHN Zhu et al. (2019), HetSANN Hong et al. (2020), HGT Hu et al. (2020), Lv et al. (2021). More related works and additional training techniques about HGNN are shown in Appendix A. However, both meta-path-based and meta-path-free methods require expensive computational overhead for message propagation and aggregation in each training epoch.

## 2.3 Scalable GNNs and HGNNs

**Scalable GNNs.** Scalable Graph Neural Networks (GNNs) are built upon the finding that the non-linear transformation is unnecessary and the weight matrices between consecutive layers can be collapsed Wu et al. (2019). Different from commonly used GNNs (i.e., GCN Kipf & Welling (2017), GraphSAGE Hamilton et al. (2017) and GAT Velickovic et al. (2018)) which adopt recursive feature propagation and non-linear transformation in each GNN layer, scalable GNNs treat the time-consuming feature propagation as a pre-processing step. Specifically, this pre-processing step can be expressed as follows:

$$\mathbf{X}^{(k)} = (\tilde{\mathbf{D}}^{r-1} \tilde{\mathbf{A}} \tilde{\mathbf{D}}^{-r})^k \mathbf{X}^{(0)} \tag{1}$$

In this equation, $\mathbf{X}^{(0)}$ represents the raw feature, and $\hat{\mathbf{A}} = \tilde{\mathbf{D}}^{r-1} \tilde{\mathbf{A}} \tilde{\mathbf{D}}^{-r}$ represents the normalized adjacency matrix $\tilde{\mathbf{A}}$, where $\tilde{\mathbf{D}}$ is the degree matrix of $\tilde{\mathbf{A}}$. After generating $\mathbf{X}^{(k)}$, the prediction results are obtained through a linear logistic regression classifier. Based on Eq. 1, several recent works have been proposed Wu et al. (2019); Zhu & Koniusz (2021); Chen et al. (2020) for scalable GNNs, which combine features at a finer granularity, i.e., hop-wise. For example, SIGN concatenates neighbor-aggregated features from different propagation layers:$[\mathbf{X}^{(0)}\mathbf{W}_0, \cdots, \mathbf{X}^{(k)}\mathbf{W}_k]$, while S$^2$GC proposes a simple spectral graph convolution to average the propagated features in different propagation layers: $\mathbf{X}^{(k)} = \sum_{i=0}^{k} \hat{\mathbf{A}}^i \mathbf{X}^{(0)}$. In contrast to these methods, GAMLP Zhang et al. (2021) further considers feature propagation from a node-wise perspective, with each node having a personalized combination of the different layers of propagated features.

**Scalable HGNNs.** Inspired by the advancements of scalable GNNs, current scalable HGNNs averagely aggregate the neighbor features of each target node in the pre-processing step and combine them in different ways during model training. In contrast to scalable GNNs, these works require the generation of path-based subgraphs to guide the propagation of graph structure information. Based on different subgraph generation methods, existing scalable HGNNs can be classified into two categories: *relation-based propagation* and *meta-path-based propagation*.

Nars Yu et al. (2020) is the relation-based propagation method that attempts to apply scalable homogeneous graph work Rossi et al. (2020) to HGNNs. Specifically, it repeatedly samples a subset of relation types to generate corresponding relation subgraphs. Then it performs neighbor averaging on these relation subgraphs, and the generated features are concatenated in a hop-wise manner, similar to SIGN. These features are then trained with a simple MLP model. Given the hop $k$, its entire procedure can be expressed as:

$$\mathbf{X}^{(n)} = \hat{\mathbf{A}}_{o_i,o_j}^n \mathbf{X}^{(0)}, n \in k \quad \mathbf{Y} = \mathrm{softmax}(\mathrm{SIGN}([\mathbf{X}^{(0)}\mathbf{W}_0, \cdots, \mathbf{X}^{(k)}\mathbf{W}_k])) \tag{2}$$

where $\mathbf{X}^{(0)} = \mathbf{X}_{o_j}$, $\mathbf{X}^{(n)}$ is the aggregated features after n-hops, , and $\hat{\mathbf{A}}_{o_i,o_j}$ is the mean aggregator. $\mathbf{W}_0, \mathbf{W}_1, \cdots, \mathbf{W}_k$ are learnable weight matrices, $\mathbf{Y}$ is the prediction result.

Unlike the relation-based propagation method, the meta-path-based propagation method SeHGNN Yang et al. (2023) repeatedly iterates over different relation types to generate general meta-paths that contain all paths within a specified hop. Same as Nars, SeHGNN also proposes the mean aggregator of neighbor features to replace the attention mechanism. Then it uses the transformer model to fuse the semantic information. Given the hop $k$, the entire procedure of SeHGNN can be expressed as:

$$\mathbf{X}^{(p)} = \hat{\mathbf{A}}_{o_t,o_1}\hat{\mathbf{A}}_{o_1,o_2}, \cdots, \hat{\mathbf{A}}_{o_{k-1},o_s}\mathbf{X}^{(0)}, p \in \Phi \quad \mathbf{Y} = \mathrm{softmax}(\mathrm{Trans}([\mathbf{X}'^{(p)}, p \in \Phi])) \tag{3}$$

where $\mathbf{X}^{(0)} = \mathbf{X}_{o_s}$, $\mathbf{X}'^{(p)}$ is $\mathbf{X}^{(p)}$ after feature projection, and each $\hat{\mathbf{A}}_{o_i,o_j}$ is the mean aggregator.

## 3 OBSERVATION AND INSIGHT

Existing HGNNs usually adopt two key techniques for scalability: fixed mean aggregator and general subgraph construction. These techniques, however, may also lead to some potential issues, such as the fixed knowledge extraction and the buried graph structure information. In this section, we conduct a quantitative analysis of these two techniques and their issues and offer insights into how HGAMLP overcomes them.

### 3.1 INSIGHT ON KNOWLEDGE EXTRACTOR

The existing scalable HGNNs uniformly employ the fixed mean aggregator for local knowledge extraction. For each target node, they assume that the difference in weight values within the same type of relationship is similar. However, in real-world graphs, the importance of neighbor features to the target node varies, and this phenomenon becomes more prominent when dealing with heterogeneous graph scenes. To investigate this problem, we use HGB (a

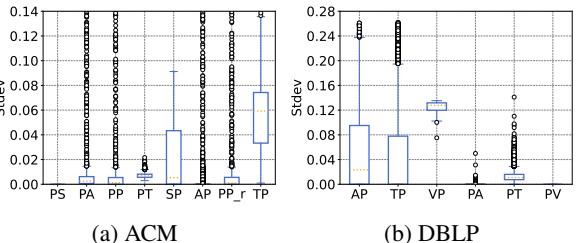

(a) ACM          (b) DBLP

Figure 3: The box plot of standard deviations of attention values for different relations in HGB. The X-axis represents different relationship types.

node-level attention-based method) to observe the attention value of all relation types in the ACM and DBLP datasets. For each type of relation, we calculate the standard deviation of attention values for each target node and illustrate it with box plots. As shown in Figure 3, the difference in attention value within the same type of relationship is variable and has many outliers. Besides, the difference in attention value in different relationships is variable, some are small and some are very large. Based on this observation, the mean aggregator can only approximately reflect the similarity of some relationship types and cannot represent the similarity of all types.

On the other hand, for global knowledge extractors, existing methods aim at combining the semantic information of all relevant meta-paths in a coarse-grained manner to obtain the final embedding. In practice, each meta-path contributes differently to each node, indicating that each node should have a personalized combination of global information. As shown in Figure 4(a), different nodes require different meta-paths to achieve high accuracy. However, current scalable approaches such as Nars and SeHGNN do not explicitly consider this issue and typically follow the well-established GNN/NLP model to do a semantic fusion of heterogeneous graphs, which may lead to sub-optimal performance.

**Insight**: Existing approaches use a fixed mean aggregator to extract local knowledge, however, the weight distribution on heterogeneous graphs is naturally uneven and this approach cannot capture richer weight distribution. Therefore, we expect to employ a local multi-knowledge extractor to enhance the express power of the captured graph structure information. Besides, to enable the node-adaptive fusion of local and global knowledge, we require a general and efficient method that can explicitly encode the importance of different knowledge present in heterogeneous graphs.

## 3.2 INSIGHT ON GENERAL SUBGRAPH CONSTRUCTION

To capture higher-order semantics, the general subgraph construction method iterates over the initial relation type, generating higher-order graph structure information through the adjacency matrix multiplication. For instance, given the relation types "Author writes Paper" (i.e., $P \leftarrow A$) and "Paper written by Author" (i.e., $A \leftarrow P$) as two first-order meta-paths, we can use $A_{PA} \times A_{AP}$ to obtain a second-order meta-path $A_{PAP}$.

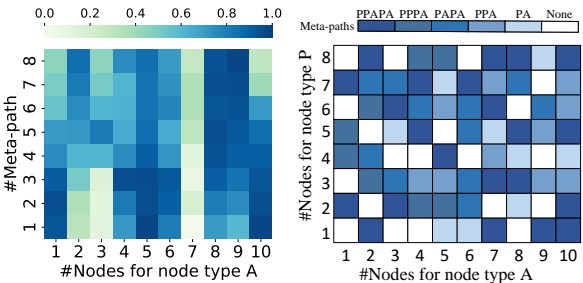

(a) Inconsistent global receptive field
(b) Redundancy of meta-path PPAPA

Figure 4: (Left) Test accuracy of SeHGNN on 10 randomly sampled nodes on DBLP. The color from green to blue represents the correct prediction ratio in 100 different runs. (Right) Path tracing of high-order meta-path PPAPA on a random sample 8×10 subgraph on ACM.

However, this approach has some limitations. As described in Section 1, when the number of hops increases in existing methods, the accuracy rate gradually decreases. We can explore the reason behind this issue by revisiting cumulative matrix multiplication. During this process, we obtain a resulting matrix where each non-zero element denotes the total number of paths that start from the source node to the destination node within up to $k$ steps. Therefore, the resulting matrix encompasses not only the paths that precisely take $k$ steps but also those that take fewer than $k$ steps.

As further explained in Figure 4(b), we utilize the relationship types $P \leftarrow P$ and $P \leftarrow A$ to conduct matrix multiplication within 4 hops on the ACM dataset, aiming to acquire the graph structure information of the meta-path PPAPA. We randomly sample a subgraph of PPAPA, containing 10 nodes of type A and 8 nodes of type P. Each cell in the figure represents the redundant information contained in each node pair of PPAPA. Specifically, for the meta-path PPAPA, starting from the rightmost starting point A can reach the leftmost end point P through various paths such as PA, PPA, PAPA, PPPA, and PPAPA. We track and compare the adjacency matrix generated by PPAPA with other low-order meta-paths from A to P. If the cells of the two compared matrices are both non-none, it means that node P can capture the information of node A through a low-order path. Moreover, PPA, PAPA, and PPPA also have the same redundancy problem. Therefore, the high-order meta-path obtained by this method contains a significant amount of redundant information from lower-order meta-paths.

**Insight**: As the propagation hop increases, numerous redundant higher-order meta-paths are generated. The previous methods generate meta-paths by iterating 1-hop relation, resulting in redundant information for higher-order meta-paths. Although they continuously enhance low-order information, they also bury valuable graph structure information in higher-order meta-paths. Consequently, a de-redundancy method is necessary to extract the structure of the high-order meta-path itself, and we propose to utilize mask matrices to achieve this goal.

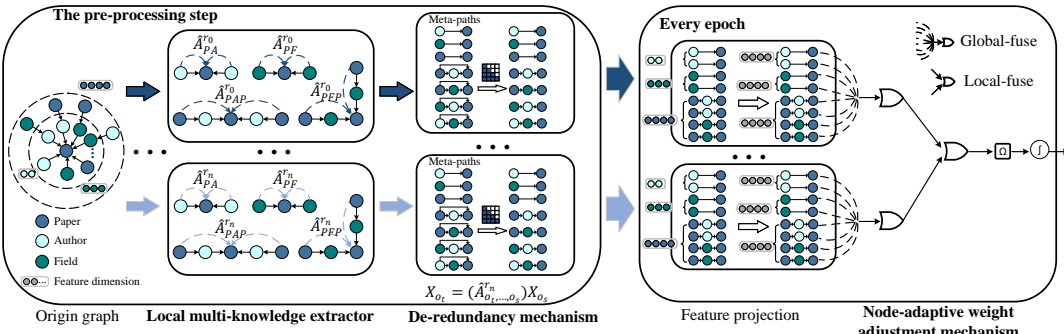

Figure 5: The architecture of HGAMLP

# 4 METHOD

This section introduces HGAMLP, a scalable and efficient framework for HGNNs as illustrated in Figure 5. It consists of three main components: a local multi-knowledge extractor, a de-redundancy mechanism, and a node-adaptive weight adjustment mechanism. The local multi-knowledge extractor enhances the node representation by using various local knowledge extractors. The de-redundancy mechanism refines graph structure information by eliminating redundant information for each meta-path. After feature projection through MLP block, the node-adaptive weight adjustment mechanism provides each node with a personalized combination of local and global knowledge simply and efficiently. The algorithm for our overall training procedures is shown in Appendix B.1.

## 4.1 LOCAL MULTI-KNOWLEDGE EXTRACTOR

The feature propagation with $k$ hops in GNNs can be represented by $\hat{\mathbf{A}}^k = (\tilde{\mathbf{D}}^{r-1}\tilde{\mathbf{A}}\tilde{\mathbf{D}}^{-r})^k$, where $\tilde{\mathbf{D}}$ is the degree matrix of $\tilde{\mathbf{A}}$, and different values of $r$ can generate different normalized adjacency matrices Kipf & Welling (2017); Xu et al. (2018); Zeng et al. (2020). Inspired by the normalization method in these previous works, normalization on heterogeneous graphs can also be expressed by setting different values of $r$. Specifically, we can set different $r$ values as corresponding knowledge extractors to capture various information from the graph structure, the formula can be expressed as:

$$\hat{\mathbf{A}}^r_{o_i,o_j} = \tilde{\mathbf{D}}^{r-1}_{o_i,o_j} A_{o_i,o_j} \tilde{\mathbf{D}}^{-r}_{o_j,o_i} \tag{4}$$

where $\tilde{\mathbf{D}}^{r-1}_{o_i,o_j}$ represents the degree matrix of the target node type $o_i$ for the meta-path $o_i, o_j$, $\tilde{\mathbf{D}}^{-r}_{o_j,o_i}$ represents the degree matrix of the source node type $o_j$ corresponding to the meta-path $o_j, o_i$. Following Eq. 4, the mean aggregator in SeHGNN Yang et al. (2023) and Nars Yu et al. (2020) can be regarded as the case of $r = 0.0$, i.e., $\hat{\mathbf{A}}^{0.0}_{o_i,o_j} = \tilde{\mathbf{D}}^{-1}_{o_i,o_j} \mathbf{A}_{o_i,o_j}$. Based on Eq. 4, the graph structure information propagation for a k-hop meta-path in scalable HGNNs can be expressed as:

$$\hat{\mathbf{A}}^r_{o_t,\cdots,o_s} = \hat{\mathbf{A}}^r_{o_t,o_1} \hat{\mathbf{A}}^r_{o_1,o_2}, \cdots, \hat{\mathbf{A}}^r_{o_{k-1},o_s} \tag{5}$$

with Eq. 4 and Eq. 5, we can capture richer local knowledge for each node. Specifically, during the pre-processing step, we utilize various knowledge extractor factors denoted as $R = r_0, r_1, \cdots, r_n$ to perform the multi-knowledge extraction operation, where $n$ represents the number of local knowledge extractors. Each local knowledge extractor employs Eq. 4 to generate the corresponding normalized weight matrices, obtaining different local knowledge. Subsequently, Eq. 5 is used to generate the adjacency matrix corresponding to each meta-path.

## 4.2 DE-REDUNDANCY MECHANISM

As mentioned earlier, scalable HGNNs typically employ the general subgraph construction method to generate graph structure information for the target meta-path. However, this approach results in a significant amount of redundant information in the generated adjacency matrix. For instance, consider the toy graph in Figure 6, which contains two node types ($p$ and $q$) and edge relations ($p \leftarrow q$ and $q \leftarrow p$). If we want to obtain the graph structure information of $p \leftarrow q \leftarrow p$, the existing subgraphs construction method (shown in Figure 6(a)) multiplies the normalized adjacency matrix $\hat{\mathbf{A}}_{pq} \times \hat{\mathbf{A}}_{qp}$ to obtain $\hat{\mathbf{A}}_{pqp}$. However, $\hat{\mathbf{A}}_{pqp}$ also includes information from $p \leftarrow p$, not just

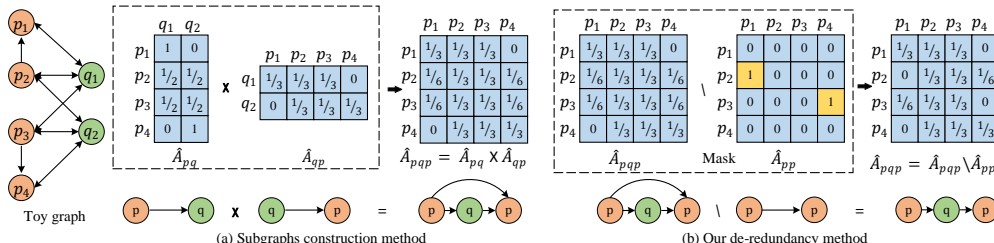

Figure 6: De-redundancy mechanism

the specified meta-path $p \leftarrow q \leftarrow p$. This redundancy worsens as the meta-path length increases. To eliminate this redundancy and obtain purely specified meta-path graph structure information, we adopt the de-redundancy method depicted in Figure 6(b). In general, we identify all lower-order meta-paths of the specified high-order meta-path and use these lower-order meta-paths as mask matrices to de-redundant (set zero) the specified high-order meta-path adjacency matrix. The algorithm for our de-redundancy mechanism is shown in Appendix B.1.

### 4.3 NODE-ADAPTIVE WEIGHT ADJUSTMENT MECHANISM

After completing the preprocessing step, each local knowledge extractor obtains multiple de-redundant global knowledge. During the model training step, we employ the node-adaptive weight adjustment mechanism to fuse the global knowledge from each local knowledge extractor, resulting in different fused local knowledge. This knowledge is further fused to obtain node embeddings.

**Adaptive Global Knowledge Fusion.** For a heterogeneous graph, it would be desirable to customize the receptive field of each node. However, existing approaches either rely on simple hop-level semantic fusion methods or node-level complex models. Unfortunately, neither approach is both simple and efficient. To address this issue, we propose a novel and efficient node-adaptive weight adjustment mechanism that can explicitly formulate a suitable receptive field for each node, thereby enhancing the expressiveness of each node. An illustration of our proposed mechanism is provided in Appendix B.2. Our approach only requires a trainable shared vector $s$ to generate a global attention pool for all nodes. For each local knowledge extractor $r$, the formulation of global knowledge fusion method is expressed as follows:

$$\mathbf{H^r} = \mathbf{h}_1^r||\mathbf{h}_2^r||\cdots||\mathbf{h}_k^r, k \in |\Phi| \qquad \mathbf{S}^r = Sigmoid(squeeze(\mathbf{H}^r\mathbf{s}))$$

$$\mathbf{Global_{att}} = \frac{exp(\mathbf{S}_i^r)}{\sum_{i=1}^{k} exp(\mathbf{S}_i^r)}, k \in |\Phi| \quad \mathbf{X}_{local}^r = \oplus_{i\in n}\mathbf{H}^r[:,i,:] \odot \mathbf{Global_{att}}[:,i], k \in |\Phi| \tag{6}$$

where $\mathbf{h}_k^r$ is the semantic vector correspond to each meta-path $k$. $\mathbf{Global}_{att}$ represents a global attention pool that stores the importance of different meta-paths (i.e., global knowledge) to each node. After that, we use different ensemble strategies $\oplus$ to fuse $\mathbf{H}^r$. For medium-scale datasets, we adopt a "weight-sum" strategy that involves summing the dot product of $\mathbf{Global}_{att}$ with $\mathbf{H}^r$ to derive the fused node embeddings. For large-scale datasets like Ogbn-mag, we adopt a "weight-concat-MLP" strategy, which concatenates the dot product of $\mathbf{Global}_{att}$ with $\mathbf{H}^r$ and passes the result through a Multi-Layer Perceptron to derive the node embeddings.

**Adaptive Local Knowledge Fusion.** Additionally, the node-adaptive weight adjustment mechanism can provide personalized local knowledge to each node. Specifically, We take the output results $\mathbf{X}_{local}^r$ as the input for local knowledge fusion. The formulation of the local knowledge fusion method is expressed as follows:

$$\mathbf{X^R} = \mathbf{X_{local}^{r_0}}||\mathbf{X_{local}^{r_1}}||\cdots||\mathbf{X_{local}^{r_n}} \quad \mathbf{W} = Sigmoid(squeeze(\mathbf{X^R}\mathbf{w}))$$

$$\mathbf{Local_{att}} = \frac{exp(\mathbf{W_i})}{\sum_{i=1}^{n} exp(\mathbf{W_i})} \qquad \mathbf{output} = \sum_{i\in n}\mathbf{X^R}[:,i,:] \odot \mathbf{Local_{att}}[:,i] \tag{7}$$

where $\mathbf{w}$ is a shared learnable vector, $\mathbf{Local_{att}}$ represents an attention pool that stores the importance of different local knowledge to each node. With this approach, each node is endowed with personalized local knowledge, enhancing its expressiveness. $\mathbf{output}$ represents the generated node embeddings for downstream tasks such as node classification, we then use MLP to generate the final embeddings. The difference between HGAMLP and other scalable methods can be found in Appendix B.3. Besides, HGAMLP can maintain high efficiency, and its time complexity analysis is in Appendix C.

Table 1: **Experiment results on the four datasets, where "-" means run out of memory.**

| | DBLP | | IMDB | | ACM | | Freebase | |
|---|---|---|---|---|---|---|---|---|
| | macro-F1 | micro-F1 | macro-F1 | micro-F1 | macro-F1 | micro-F1 | macro-F1 | micro-F1 |
| RGCN | $91.52 \pm 0.50$ | $92.07 \pm 0.50$ | $58.85 \pm 0.26$ | $62.05 \pm 0.15$ | $91.55 \pm 0.74$ | $91.41 \pm 0.75$ | $46.78 \pm 0.77$ | $58.33 \pm 1.57$ |
| HAN | $91.67 \pm 0.49$ | $92.05 \pm 0.62$ | $57.74 \pm 0.96$ | $64.63 \pm 0.58$ | $90.89 \pm 0.43$ | $90.79 \pm 0.43$ | $21.31 \pm 1.68$ | $54.77 \pm 1.40$ |
| RSHN | $93.34 \pm 0.58$ | $93.81 \pm 0.55$ | $59.85 \pm 3.21$ | $64.22 \pm 1.03$ | $90.50 \pm 1.51$ | $90.32 \pm 1.54$ | - | - |
| HetGNN | $91.76 \pm 0.43$ | $92.33 \pm 0.41$ | $48.25 \pm 0.67$ | $51.16 \pm 0.65$ | $85.91 \pm 0.25$ | $86.05 \pm 0.25$ | - | - |
| MAGNN | $93.28 \pm 0.51$ | $93.76 \pm 0.45$ | $56.49 \pm 3.20$ | $64.67 \pm 1.67$ | $90.88 \pm 0.64$ | $90.77 \pm 0.65$ | - | - |
| HetSANN | $78.55 \pm 2.42$ | $80.56 \pm 1.50$ | $49.47 \pm 1.21$ | $57.68 \pm 0.44$ | $90.02 \pm 0.35$ | $89.91 \pm 0.37$ | - | |
| HGT | $93.01 \pm 0.23$ | $93.49 \pm 0.25$ | $63.00 \pm 1.19$ | $67.20 \pm 0.57$ | $91.12 \pm 0.76$ | $91.00 \pm 0.76$ | $29.28 \pm 2.52$ | $60.51 \pm 1.16$ |
| HGB | $94.01 \pm 0.24$ | $94.46 \pm 0.22$ | $63.53 \pm 1.36$ | $67.36 \pm 0.57$ | $93.42 \pm 0.44$ | $93.35 \pm 0.45$ | $47.72 \pm 1.48$ | $66.29 \pm 0.45$ |
| SeHGNN | $94.86 \pm 0.14$ | $95.24 \pm 0.13$ | $66.63 \pm 0.34$ | $68.21 \pm 0.32$ | $93.95 \pm 0.48$ | $93.87 \pm 0.50$ | $50.71 \pm 0.44$ | $63.41 \pm 0.47$ |
| HGAMLP | $\mathbf{95.49 \pm 0.11}$ | $\mathbf{95.83 \pm 0.08}$ | $\mathbf{67.17 \pm 0.29}$ | $\mathbf{69.40 \pm 0.46}$ | $\mathbf{94.53 \pm 0.12}$ | $\mathbf{94.46 \pm 0.10}$ | $\mathbf{51.70 \pm 0.35}$ | $\mathbf{66.57 \pm 0.21}$ |

Table 2: **Experiment results on the Ogbn-mag compared with other methods on the OGB leaderboard, using "emb" for extra embeddings and "ms" for multi-stage training.**

| Methods | Val acc | Test acc | Trick | Val acc | Test acc |
|---|---|---|---|---|---|
| SAGN | $52.25 \pm 0.30$ | $51.17 \pm 0.32$ | | $55.91 \pm 0.17$ | $54.40 \pm 0.15$ |
| GAMLP | $53.23 \pm 0.23$ | $51.63 \pm 0.22$ | +emb | $57.02 \pm 0.41$ | $55.90 \pm 0.27$ |
| SeHGNN | $55.95 \pm 0.11$ | $53.99 \pm 0.18$ | +ms | $59.17 \pm 0.09$ | $57.19 \pm 0.12$ |
| PSHGCN | $56.16 \pm 0.21$ | $54.57 \pm 0.16$ | | $59.43 \pm 0.15$ | $57.52 \pm 0.11$ |
| HGAMLP | $\mathbf{56.32 \pm 0.06}$ | $\mathbf{54.61 \pm 0.10}$ | | $\mathbf{59.60 \pm 0.11}$ | $\mathbf{57.60 \pm 0.10}$ |

# 5 EXPERIMENTS

## 5.1 EXPERIMENT SETTINGS

**Datasets**. Four middle-scale datasets from the HGB benchmark and a large-scale dataset Ogbn-mag from OGB Challenge Hu et al. (2021) are used for the node classification task.

**Baselines and Settings**. For middle-scale datasets, we compare HGAMLP with RGCN, HetGNN, HAN, MAGNN, RSHN, HetSANN, HGT, HGB, and the SOTA scalable HGNN SeHGNN. All results are repeated five times and report the mean performance and the standard deviations. For large-scale datasets, we compare the methods SAGN, GAMLP, SeHGNN, and PSHGCN He et al. (2023) on the Ogbn-mag leaderboard. To ensure fairness, we follow the rules of the Ogbn-mag leaderboard run HGAMLP 10 times, and report the average performance. Datasets properties and setup details are shown in Appendix D. The source code of HGAMLP can be found in `https://anonymous.4open.science/r/HGAMLP_ICLR_2024/`.

## 5.2 EXPERIMENT RESULTS

**Experiments on HGB Benchmark.** We present the evaluation results of our node classification predictions on four middle-scale heterogeneous graph datasets. Considering that only SeHGNN utilizes the label propagation trick, we compare the version of SeHGNN that does not employ label propagation in the experiment. Table 1 shows that HGAMLP method outperforms all advanced HGNN methods on all the datasets. Notably, the previous work on the Freebase dataset fails to maintain high accuracy in both macro-F1 and micro-F1 simultaneously. In contrast, HGAMLP maintains the highest accuracy in both micro-F1 and macro-F1.

**Experiments on Ogbn-mag.** Achieving good prediction performance on large-scale graphs is a challenging task in HGNN. To evaluate the scalability of HGAMLP, we test it on a large-scale heterogeneous graph dataset Ogbn-mag. Due to the lack of node features and unbalanced data distribution in the Ogbn-mag, existing methods often rely on extra embedding and multi-stage training techniques (more details in Appendix A.2). To ensure a fair comparison, we conduct ablation experiments with and without these tricks. Table 2 shows that HGAMLP outperforms other baselines with and without tricks. Our excellent performance on Ogbn-mag stems from our node-adaptive weight adjustment mechanism, which highlights the superiority of our semantic fusion model.

**Efficiency Analysis.** To further demonstrate the exceptional efficiency of HGAMLP, we conducted a comprehensive comparison with previous HGNNs. Figure 7 compares the total training time using the early-stop strategy on the DBLP dataset. For HGAMLP, SeHGNN, and Nars, we take the precomputation time into account when measuring the total training time. Figure 7 reflects that HGAMLP offers faster training speed and higher accuracy than existing state-of-the-art methods.

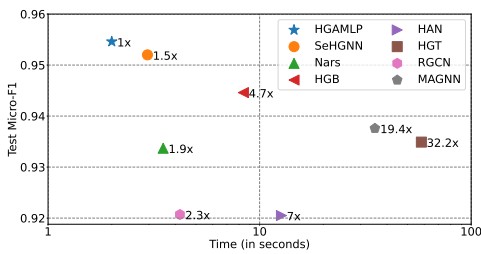
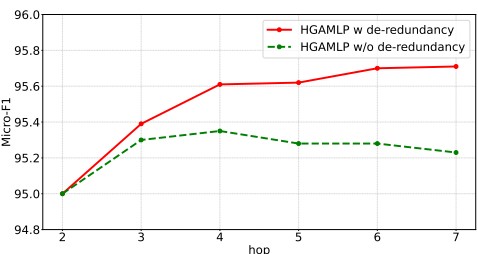

Figure 7: Performance over training time on DBLP.

Figure 8: Ablation study to verify the effectiveness of de-redundancy mechanism.

Table 3: **Ablation study to verify the effectiveness of local multi-knowledge extractors.**

|  |  | r=0.0 | r=0.1 | r=0.2 | r=0.3 | r=0.4 | r=0.5 | $r_{fuse}$ |
|---|---|---|---|---|---|---|---|---|
| ACM | macro-F1 | $94.40 \pm 0.14$ | $94.31 \pm 0.19$ | $94.25 \pm 0.24$ | $94.18 \pm 0.20$ | $94.03 \pm 0.32$ | $94.19 \pm 0.17$ | $\mathbf{94.53 \pm 0.12}$ |
|  | micro-F1 | $94.37 \pm 0.14$ | $94.23 \pm 0.19$ | $94.17 \pm 0.25$ | $94.37 \pm 0.20$ | $93.96 \pm 0.33$ | $94.11 \pm 0.17$ | $\mathbf{94.46 \pm 0.10}$ |
| DBLP | macro-F1 | $95.18 \pm 0.12$ | $95.17 \pm 0.19$ | $95.21 \pm 0.32$ | $94.98 \pm 0.23$ | $94.88 \pm 0.25$ | $94.84 \pm 0.36$ | $\mathbf{95.49 \pm 0.16}$ |
|  | micro-F1 | $95.53 \pm 0.11$ | $95.51 \pm 0.17$ | $95.54 \pm 0.28$ | $95.37 \pm 0.20$ | $95.25 \pm 0.23$ | $95.22 \pm 0.32$ | $\mathbf{95.83 \pm 0.15}$ |

Table 4: **Ablation study on the effectiveness of node-adaptive weight adjustment mechanism.**

|  | ACM | | DBLP | |
|---|---|---|---|---|
|  | macro-F1 | micro-F1 | macro-F1 | micro-F1 |
| mean | $93.72 \pm 0.25$ | $93.63 \pm 0.26$ | $94.89 \pm 0.21$ | $95.13 \pm 0.18$ |
| tf | $94.21 \pm 0.30$ | $94.15 \pm 0.24$ | $95.06 \pm 0.14$ | $95.31 \pm 0.17$ |
| ours | $\mathbf{94.49 \pm 0.15}$ | $\mathbf{94.42 \pm 0.10}$ | $\mathbf{95.38 \pm 0.16}$ | $\mathbf{95.61 \pm 0.15}$ |

**Ablation Study.** We conduct ablation studies on the ACM and DBLP datasets to thoroughly analyze the effectiveness of our three essential components: local multi-knowledge extractors, de-redundancy mechanism, and node-adaptive weight adjustment mechanism.

*Different Local Knowledge Extractors.* We use different local knowledge extractors by varying the $r$ values. Table 3 shows that the performance varies across datasets for different $r$ values, indicating that each local knowledge extractor captures different knowledge. To enrich each node with more knowledge, we adaptively fuse the local multi-knowledge, creating fusion knowledge denoted as $r_{fuse}$. Consequently, this approach yields better results compared to using a single extractor.

*Different Subgraph Construction Methods.* We compare HGAMLP with the de-redundancy method ("HGAMLP w de-redundancy") and with the general subgraph construction method ("HGAMLP w/o de-redundancy") and evaluate the accuracy trend. As depicted in Figure 8, "HGAMLP w de-redundancy" consistently outperforms "HGAMLP w/o de-redundancy" in accuracy, and gradually improves as the number of hops increases. This observation validates the redundancy issue in the previous general subgraph construction method, which hinders the effective utilization of high-order meta-paths. Further experimental explanation of redundancy can be found in Appendix D.3.

*Different Knowledge Fusion Models.* To validate the effectiveness of our node-adaptive weight adjustment mechanism for global knowledge fusion, we compared our method "Ours" with the "mean" model (direct averaging of semantic information) and the "tf" model (Transformer model). Table 4 demonstrates that our method outperforms the others, highlighting the simplicity and effectiveness of our approach, which uses only one learnable vector for improved semantic fusion.

# 6 CONCLUSION

This paper presents HGAMLP, a novel, scalable, and efficient method for learning heterogeneous graph representations. HGAMLP refines graph structure information through the de-redundancy mechanism and captures richer graph structure information using local multi-knowledge extractors. It incorporates a node-adaptive weight adjustment mechanism that assigns varying attention values to semantic information associated with each node. This mechanism is also applicable to the node-level fusion of local multi-knowledge. Finally, HGAMLP achieves excellent results using a simple Multi-Layer Perceptron. Experimental results on node classification demonstrate that HGAMLP outperforms state-of-the-art HGNNs in terms of efficiency and accuracy.

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

The appendix is organized as follows:

**A.** More related works.

**B.** More details about HGAMLP.

**C.** Time complexity analysis.

**D.** Experiment details.

## A   MORE RELATED WORKS

### A.1   HETEROGENEOUS GRAPH NEURAL NETWORKS

**Meta-path-based Methods.** Meta-path-based HGNNs use predefined paths that specify the type of relationships between nodes to guide the message-passing process between nodes. Specifically, they first aggregate neighbor features along the meta-path and then fuse the semantic information generated by different meta-paths to obtain the final node embedding. RGCN Schlichtkrull et al. (2018) is an extension of GCN that handles graphs with multiple edge types. It applies the GCN framework to relational data modeling, and support link prediction and entity classification tasks. HetGNN Zhang et al. (2019) utilizes neighbors of varying hop distances by employing a random walk with restart strategy to sample a fixed-sized group of neighbors, which are then grouped by node type. Next, a novel architecture is utilized to aggregate feature information from the sampled neighbors. HAN Wang et al. (2019) utilizes meta-path to incorporate both node-level and semantic-level attention. It first uses node-level attention to learn the attention values for each neighbor along the meta-path. Then it uses semantic-level attention to learn the attention values of different meta-paths. The outputs are fused for node classification tasks. MAGNN Fu et al. (2020) introduces a novel meta-path-aggregated graph neural network for heterogeneous graph embedding. It devises multiple candidate encoder functions for extracting information from meta-path instances. However, both meta-path-based and meta-path-free methods require expensive computational overhead for message propagation and aggregation in each training epoch.

**Meta-path-free Methods.** Meta-path-free HGNNs do not use predefined meta-paths for message propagation. Instead, they learn to combine the embeddings of node and edge types in the graph. RSHN Zhu et al. (2019) proposes a unified model that integrates both graph and its coarsened line graph, allowing for the embedding of both nodes and edges in heterogeneous graphs, without necessitating any prior knowledge such as meta-path. HetSANN Hong et al. (2020) directly leverages and explores the structures in the heterogeneous graph to achieve more informative representations without pre-defined meta-paths. Besides, it employs a multi-layer GAT network with type-specific score functions to generate attention for different relations. HGT Hu et al. (2020) combines node and edge types using a transformer model and employs the relative temporal encoding (RTE) technique to model the dynamic dependencies. Furthermore, a mini-Batch graph sampling algorithm has been designed to support large-scale datasets. Lv et al. (2021) proposes HGB, which presents a systematical reproduction of recent HGNNs and identifies some neglected issues. Furthermore, it adopts a multi-layer GAT network as the backbone and incorporates both node features and learnable edge-type embeddings for attention generation. These methods exhibit promising outcomes for graph embedding with heterogeneous information and have the potential to advance the field of graph neural networks.

### A.2   ADDITION TRAINING TECHNIQUES

**Node embedding.** Not every node type in heterogeneous graph datasets has features. For example, in the Ogbn-mag dataset, only the "paper" type has node features. To improve prediction results, various existing methods attempt to assign features to featureless node types. These methods include padding with zeros, using meta-path2vec, and using unsupervised graph embedding methods to pre-generate features. The method of padding with zeros directly adds zero elements at the end of the feature of each node type and forcibly aligns the feature dimensions of all node types. But this approach doesn't perform well. Some HGNNs, such as RHGNN Yu et al. (2021) and LEGNN Yu et al. (2022), utilize the second method. Nars uses the third method to generate TransE relational graph

embeddings, which perform better than the first two methods. Similarly, SeHGNN and GAMLP follow the third method to generate Complex embeddings and have achieved good results.

**Label propagation.** To fully utilize the labels in the training set, homogeneous graph-based research Shi et al. (2021); Huang et al. (2021); Sun & Wu (2021) utilize them as additional inputs to improve model performance. Specifically, the label embedding matrix $\hat{\mathbf{Y}}^{(0)}$ is initialized to zeros and then populated with the hard training labels $\mathbf{Y}_{v_l}$, converted to one-hot format $\hat{\mathbf{Y}}_{v_l}^{(0)}$. This populated matrix is propagated alongside the normalized adjacency matrix $\hat{A}$.

$$\hat{\mathbf{Y}}_{v_l}^{(0)} = \mathbf{Y}_{v_l}, \hat{\mathbf{Y}}^{(k+1)} = \hat{\mathbf{A}}\hat{\mathbf{Y}}^{(k)} \tag{8}$$

where $v_l$ is the labeled node-set, and $k$ is the number of steps for label propagation. A similar method is used for heterogeneous graphs Yang et al. (2023), which employs meta-paths to perform label propagation. Since the propagated labels are of the target node type, constructing a label-based meta-path necessitates the source node and the target vertex to be of the same vertex type. A potential problem with adopting label propagation is that the trained nodes may use their own real labels, resulting in data leakage. To address this issue, SeHGNN removes diagonal values after matrix multiplication. Given a meta-path $p = o_t, \cdots, o_s$, the formula is expressed as follows:

$$\mathbf{Y}^p = rm\_diag(\hat{\mathbf{A}}^p)\hat{\mathbf{Y}}_{v_l}^{(0)}, \hat{\mathbf{A}}^p = \hat{\mathbf{A}}_{o_t,o_1}\hat{\mathbf{A}}_{o_1,o_2}\cdots\hat{\mathbf{A}}_{o_{k-1},o_s} \tag{9}$$

**Feature projection.** Due to the heterogeneity of the graph, features of different node types may have varying dimensions. Thus, recent HGNNs utilize methods to project different feature dimensions of various node types into the same space. In previous HGNNs, features are projected during the training epoch. HAN Wang et al. (2019) achieves this by designing a type-specific weight matrix to project different types of nodes' features into the same feature space. On the other hand, HGT Hu et al. (2020) and HGB Lv et al. (2021) use a linear layer with a bias for each node type. For scalable HGNNs, such as Nars Yu et al. (2020), the features of different subgraphs are directly added during the pre-processing step. Therefore, it is crucial to ensure that the feature dimensions are consistent beforehand. To achieve this, Nars employs several uniform distribution matrices to randomly project the features into the same dimension. However, this method lacks interpretability and fails to train node embeddings effectively. To address this issue, SeHGNN Yang et al. (2023) generates different meta-paths during the pre-processing step, each with varying feature dimensions. At the beginning of each training epoch, different linear layers are utilized to project the feature dimensions into the same space, allowing node embeddings to continuously learn better expressions during training.

**Multi-stage training.** Many real-world graphs often have a limited number of labels, making it challenging to effectively propagate supervision information throughout the graph. To address this issue, recent studies Sun et al. (2020), Zhang et al. (2021) drew inspiration from self-training You et al. (2020) and extended the method to multi-stage training. The approach involves adding nodes with high-confidence model predictions to the labeled training set at each stage and retraining the model in new stages, thus continuously expanding the utilization of supervised information throughout the graph. GAMLP, SAGN, and SeHGNN have used this method on the large-scale heterogeneous graph dataset Ogbn-mag to achieve excellent results.

## B MORE DETAILS ABOUT HGAMLP

### B.1 ALGORITHMS

Algorithm 1 illustrates our overall training procedure through pseudo-code. In the pre-processing step, HGAMLP first utilizes different local knowledge extractors $r$ to capture different local knowledge for each meta-path and generates its adjacency matrix $\hat{\mathbf{A}}_{o_t,\cdots,o_s}^r$ (line 1-4 in Algorithm 1). After that, HGAMLP utilizes the de-redundancy mechanism to eliminate redundant graph structure information for each meta-path, generating the de-redundancy matrices (lines 5-7 in Algorithm 1). Each $\hat{\mathbf{A}}_{o_t,\cdots,o_s}^r$ generated by the de-redundancy mechanism is then used to aggregate neighbor features to generate global knowledge (line 8-9 in Algorithm 1). In the model training step, each local knowledge extractor employs an MLP block to project each global knowledge into the same data space during feature projection (lines 18-19 in algorithm 1). Then the node adaptive weight adjustment mechanism first fuses all the global knowledge $\mathbf{h}^r$ under each local extractor to generate $\mathbf{X}_{local}^r$

---

**Algorithm 1** The overall training process of HGAMLP.

---

**Input:**

    Raw features of source node type: $\mathbf{X}_{o_s}$

    Raw labels: $\mathbf{Y}$

    Meta-path list: $\Phi$

    Adjacency matrices: $\{\mathbf{A}_{o_i,o_j} : o_i, o_j \in T_v\}$

    Local knowledge extractors: $R$

**Output:** Node classification prediction results $Pred$ of target node type $o_t$

1:  **% Local Knowledge Extractor**

2:  **for** $r \in R$ **do**

3:     **for** each $o_t, \cdots, o_s \in \Phi$ **do**

4:        $\hat{\mathbf{A}}^r_{o_t,\cdots,o_s} = \hat{\mathbf{A}}^r_{o_t,o_1} \hat{\mathbf{A}}^r_{o_1,o_2}, \cdots, \hat{\mathbf{A}}^r_{o_{k-1},o_s}$

5:        **%De-redundancy Mechanism**

6:        $\hat{\mathbf{A}}^r_{o_t,\cdots,o_s} = De\text{-}redundancy(\hat{\mathbf{A}}^r_{o_t,\cdots,o_s}, \hat{\mathbf{A}}_{pool})$

7:        $\hat{A}_{pool}.append(\hat{\mathbf{A}}^r_{o_t,\cdots,o_s})$

8:        %Neighbor aggregation

9:        $\mathbf{X}^r_{o_t,\cdots,o_s} = \hat{\mathbf{A}}^r_{o_t,\cdots,o_s} \mathbf{X}_{o_s}$

10:     **end for**

11:     %Collect all feature aggregation matrices

12:     $\mathbf{X}^r = \{\mathbf{X}^r_{o_t,\cdots,o_s} : o_t, \cdots, o_s \in \Phi\}$

13: **end for**

14: %Collect local multi-knowledge extractor

15: $Q = \{\mathbf{X^r} : r \in R\}$

16: **for** each epoch **do**

17:     **for** $\mathbf{X^r} \in Q$ **do**

18:     % feature projection

19:     $\mathbf{h}^r = \{MLP(\mathbf{X}^r_{o_t,\cdots,o_s}) : for\ \mathbf{X}^r_{o_t,\cdots,o_s} \in \mathbf{X}^r\}$

20:     **%Node-adaptive Weight Adjustment Mechanism**

21:     %Fuse global knowledge

22:     $\mathbf{X}^r_{local} = Adaptive\text{-}Global\text{-}Knowledge\text{-}Fusion(\mathbf{h}^r)$

23:     **end for**

24:     $\mathbf{X}^R_{local} = \{\mathbf{X}^r_{local} : r \in R\}$

25:     %Fuse local knowledge

26:     $\mathbf{Z}_{final} = Adaptive\text{-}Local\text{-}Knowledge\text{-}Fusion(\mathbf{X}^R_{local})$

27:     % Node classification task

28:     $\mathbf{Pred} = MLP(\mathbf{Z}_{final})$

29:     Calculate loss function $\mathbf{L}(\mathbf{y}, \mathbf{Pred}) = -\sum_{i=1}^{n} y_i \log \mathbf{pred}_i$

30:     back-propagation

31: **end for**

32: **return Pred**

---

---

**Algorithm 2** De-redundancy Mechanism.

---

**Input:**

  $\hat{\mathbf{A}}_{pool}$ :A de-redundancy adjacency matrix storage pool

  $\hat{\mathbf{A}}_{o_t,\cdots,o_s}$: Adjacency matrix to be de-redundant

**Output:** $\hat{\mathbf{A}}_{o_t,\cdots,o_s}$
 1: % generate sub-meta-path
 2: $sub\text{-}meta\text{-}path = DFS(o_t,\cdots,o_s)$
 3: **for** $name \in sub\text{-}meta\text{-}path$ **do**
 4:   **if** $name \in \hat{\mathbf{A}}_{pool}.keys()$ **then**
 5:     $\hat{\mathbf{A}}_{o_t,\cdots,o_s} = \hat{\mathbf{A}}_{o_t,\cdots,o_s}\backslash\hat{\mathbf{A}}_{pool}[name])$
 6:   **else**
 7:     continue
 8:   **end if**
 9: **end for**
10: **return** $\hat{\mathbf{A}}_{o_t,\cdots,o_s}$

---

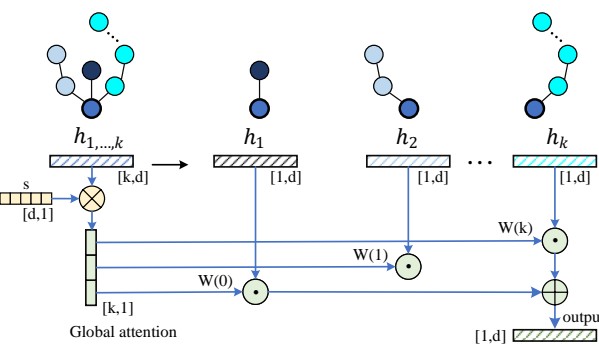

Figure 9: Node-adaptive weight adjustment mechanism, where $h_{1,\dots,k}$ represents the concatenation of features corresponding to k metapaths.

(lines 21-22 in Algorithm 1). After that, all local knowledge $\mathbf{X}^R_{local}$ is further fused to generate the node embedding (lines 25-26 in Algorithm 1) for node classification prediction.

Algorithm 2 demonstrates our de-redundancy mechanism. For every $\hat{\mathbf{A}}_{o_t,\cdots,o_s}$ generated through the local multi-knowledge extractor, we consider it as the target redundant matrix and its meta-path as the target meta-path. Specifically, we employ a de-redundancy matrix pool denoted as $\hat{\mathbf{A}}_{pool}$ to store all normalized adjacency matrices that have undergone the de-redundancy process. As the 1-hop meta-path lacks redundancy, we initialize $\hat{\mathbf{A}}_{pool}$ with the normalized adjacency matrices of the 1-hop meta-path. We initiate the process by utilizing the DFS algorithm to traverse the target meta-path and generate all sub-meta-paths (line 2 in Algorithm 2). For each sub-meta-path, we verify its existence in $\hat{\mathbf{A}}_{pool}$. If it exists, we extract pure information using the de-redundancy mechanism (lines 4-5 in Algorithm 2). If not, we move on to the next sub-meta-path (line 7 in Algorithm 2). We repeat this process until all sub-meta-paths associated with the target meta-path have been refined. The resulting $\hat{\mathbf{A}}_{o_t,\cdots,o_s}$ is then stored in $\hat{\mathbf{A}}_{pool}$, and we proceed to eliminate redundancy in the next adjacency matrix.

### B.2 Illustration of Node-adaptive weight adjustment mechanism

The illustration of the node-adaptive weight adjustment mechanism is depicted in Figure 9.

### B.3 Relation with current methods

**HGAMLP vs. SeHGNN**. Both HGAMLP and SeHGNN employ general subgraph construction methods based on meta-paths. However, SeHGNN ignores the redundancy of high-order meta-paths, whereas HGAMLP incorporates a de-redundancy mechanism to eliminate redundancy in high-order meta-paths. In the neighbor aggregation step, SeHGNN aggregates limited knowledge using a fixed

Table 5: **Theoretical complexity of HGAMLP, Nars, GAMLP and SeHGNN in every training mini-batch.**

| Model | Featrue projection | Semantic fusion | Total |
|---|---|---|---|
| Nars | - | $\mathcal{O}(ND(mD))$ | $\mathcal{O}(ND(mD))$ |
| GAMLP | - | $\mathcal{O}(ND(mD))$ | $\mathcal{O}(ND(mD))$ |
| SeHGNN | $\mathcal{O}(NKD^2)$ | $\mathcal{O}(ND(K^2 + KD))$ | $\mathcal{O}(ND(K^2 + KD))$ |
| Ours | $\mathcal{O}(NKrD^2)$ | $\mathcal{O}(NDrK)$ | $\mathcal{O}(ND(rKD))$ |

mean aggregator, while HGAMLP utilizes the local multi-knowledge fusion method to enhance the expressive ability of each node. In terms of model selection, SeHGNN employs a transformer model, whereas HGAMLP supports both local and global knowledge fusion through a node adaptive weight adjustment mechanism, which is efficient and simple.

**HGAMLP vs. Nars**. Similar to SeHGNN, Nars also employs a general subgraph construction method and mean aggregator. However, Nars is based on relationships rather than meta-paths, which prevents explicit handling of heterogeneous feature types. Additionally, Nars inherits the SIGN model Rossi et al. (2020) for model training without considering the importance of each meta-path to each node.

**HGAMLP vs. GAMLP**. GAMLP is a scalable graph framework that integrates Nars to support heterogeneous graph processing. While GAMLP allows node-level personalized combination, it is limited by Nars and can only perform this operation at the hop level, rather than at a finer-grained meta-path level.

## C  TIME COMPLEXITY ANALYSIS

**Time analysis.** We conduct a theoretical analysis of the time complexity for Nars, GAMLP, Se-HGNN, and HGAMLP. To facilitate the analysis, we make the following assumptions: The maximum hop of the above methods is set to $m$ to ensure the same receptive field and $k$ meta-paths are generated accordingly. The input feature dimension and hidden layer dimension are set to $D$, and the number of target nodes is denoted as $N$. The parameter $r$ indicates the number of the local multi-knowledge aggregator. We set $r = 1$ for comparison because the accuracy of HGAMLP for all datasets can already exceed the existing SOTA baselines. When $r > 1$, the accuracy of our method will be further improved, while the computational overhead will increase linearly. For the pre-processing step, Nars, GAMLP, SeHGNN, and HGAMLP all employ the parameter-free approach for neighbor aggregation, which eliminates the computational overhead of neighbor aggregation in each training epoch. For the model training step, the time complexity of HGAMLP (as shown in Table 5) is significantly lower than SeHGNN, attributed to the node adaptive weight adjustment mechanism. Nars has the lowest time complexity by employing feature projection in the pre-processing step. However, this method lacks interpretability and exhibits low accuracy.

## D  EXPERIMENTAL DETAILS

### D.1  DATASET DESCRIPTION

Four middle-scale datasets from the HGB benchmark are used in our experiments and a large-scale dataset Ogbn-mag from OGB Challenge Hu et al. (2021). Table 6 shows the graph structure properties of these five datasets.

- **Middle-scale dataset**. ACM and DBLP are academic networks, the former is a citation network, and the latter is a bibliography website of computer science. IMDB is a website that contains movies and related media. Freebase is a large knowledge graph.
- **Large-scale dataset.** Ogbn-mag is a heterogeneous academic network extracted from Microsoft Academic Graph (MAG), consisting of papers (P), authors (A), fields (F), and insti-

Table 6: **Overview of the Datasets.**

| Dataset | Nodes | Nodes types | Edges | Classes |
|---------|-------|-------------|-------|---------|
| DBLP | 26128 | A,P,T,V | 239566 | 4 |
| ACM | 10942 | A,P,C,K | 547872 | 3 |
| IMDB | 21420 | A,M,D,K | 86642 | 5 |
| Freebase | 180098 | 0-7 | 1057688 | 7 |
| Ogbn-mag | 1939743 | P,F,A,I | 21111007 | 349 |

Table 7: **The maximum hops of metapaths for feature propagation and label propagation.**

| | Max Hop | Metapaths | Max Hop | Metapaths |
|---------|---------|-----------|---------|-----------|
| DBLP | 4 | 17 | - | - |
| ACM | 5 | 84 | - | - |
| IMDB | 5 | 52 | - | - |
| Freebase | 2 | 73 | - | - |
| Ogbn-mag | 2 | 10 | 2 | 5 |

tutions (I). The papers are published on 349 different venues and each paper uses word2vec to generate corresponding features.

## D.2 EXPERIMENT SETTINGS

For the medium-scale datasets, we follow the HGB benchmark that split the node labels into 24%/6%/70% for training, validation, and testing. To evaluate the effectiveness of our approach, we compare our experimental results with the baselines reported in the HGB benchmark and Se-HGNN paper. All results are repeated five times and report the mean performance and the standard deviations. For the large-scale dataset Ogbn-mag, we follow the Ogb division method according to the years, employing data before 2018 as the training set, data from 2018 as the validation set, and data after 2019 as the test set. We use the Ogb model evaluator to get validation and test accuracy (micro-F1). To ensure fairness, we follow the rules of the Ogbn-mag leaderboard run HGAMLP 10 times, and report the average performance. Considering that the SOTA methods on the Ogbn-mag leaderboard utilize additional techniques such as generative features (TransE, ComplexE) and multi-stage training Yang et al. (2023); Yu et al. (2020); Zhang et al. (2021); Sun et al. (2020); Hu et al. (2020).

We train our models using Adam optimizer Kingma & Ba (2015) during training. The learning rate is 0.001 for all middle-scale datasets and 0.002 for the large-scale dataset. HGAMLP adopts two-layer MLPs for each meta-path in feature projection, and another two-layer MLP for the outputs. The dimension of hidden vectors is 64 for ACM and DBLP, 512 for IMDB, Freebase, and Ogbn-mag. Table 7 shows the number of hops and meta-paths we use for feature propagation and label propagation on each dataset.

For the training budget, we train every middle-scale dataset with 200 epochs and we terminate the training process if the validation accuracy does not improve for 20 consecutive steps. For large-scale datasets, i.e., Ogbn-mag, we use multi-stage training techniques, and the training budget of each stage is set to 400, 400, 400, and 500 and we terminate the training process if the validation accuracy does not improve for 100 consecutive steps.

The experiments are conducted on a machine with Intel(R) Xeon(R) Gold 5120 CPU @ 2.20GHz and a single TITAN RTX GPU with 24GB GPU memory. The operating system of the machine is Ubuntu 16.04. As for software versions, we use Python 3.7, Pytorch 1.12.1, and CUDA 10.1.

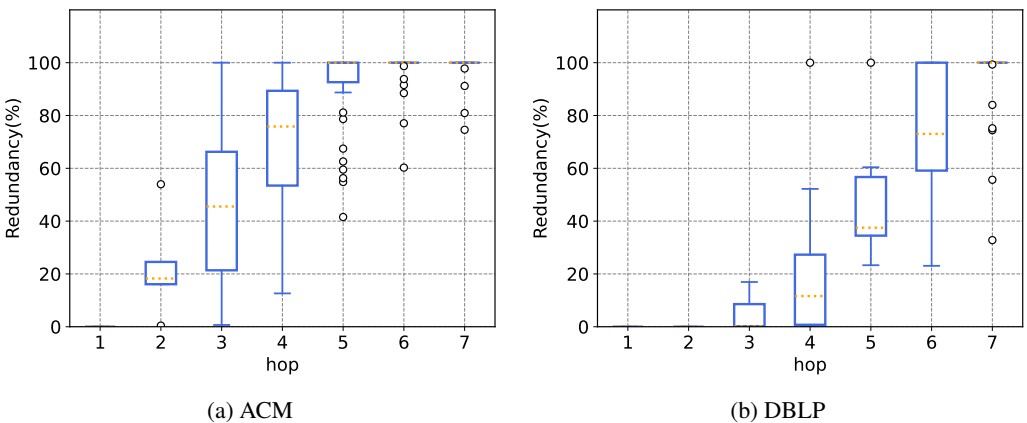

(a) ACM                  (b) DBLP

Figure 10: The box plots of the redundancy ratio for different meta-paths ranges from 1 to 7 hops on the ACM and DBLP datasets.

### D.3 MORE ABLATION STUDY

In order to facilitate the redundancy problem more intuitively, we calculate the proportion of redundancy for all meta-paths generated by each hop in the ACM and DBLP datasets and illustrate it with box plots. Specifically, we generate all meta-paths ranging from 1 to 7 hops and record the redundancy ratio for each meta-path. For the ACM dataset, the number of meta-paths generated by each hop is [1,3,3,9,9,27,27], while for the DBLP dataset, the number of meta-paths generated by each hop is [3,5,11,21,43,85,171]. As illustrated in Figure 10, it is evident that the redundancy of meta-paths increases significantly with the number of hops. Notably, in the ACM and DBLP datasets, the redundancy of most meta-paths reaches 100% when the hop value is set to 7.

