# OpenReview forum: "HGAMLP: A Scalable Training Framework for Heterogeneous Graph Learning"
_ICLR.cc/2024/Conference — ICLR 2024 Conference Withdrawn Submission_

### Official Review · Reviewer_Qrvt · 2023-10-19

**Soundness:** 2 fair
**Presentation:** 3 good
**Contribution:** 2 fair
**Rating:** 3
**Confidence:** 4

**Summary:**

This work presents an efficient HGNN model named HGAMLP which aims to better fuse local and global knowledge.

**Strengths:**

- This work is easy to follow.
- The provided supplemental materials facilitate good reproducibility of this work.
- The experiments are extensive, and the results are promising.

**Weaknesses:**

Strengths*
- This work is easy to follow.
- The provided supplemental materials facilitate good reproducibility of this work.
- The experiments are extensive, and the results are promising.


Weaknesses*
- Some arguments are not convincing. Examples are as follows.

	(1) The authors claim that existing HGNN methods adopt a fixed knowledge extractor. However, the multihead attention for node-level aggregation in [1,3] can also be viewed as various knowledge extractors.

	(2) The authors claim that existing HGNN methods bury the graph structure information of the higher-order meta-paths and fail to fully leverage the higher-order global information. However, previous methods [2,4] are able to automatically discover any-order of meta-paths and effectively exploit the structure information conveyed by the discovered meta-paths.

	(3) The authors claim that scaling them to large graphs is challenging due to the high computational and storage costs of feature propagation and attention mechanisms. However, the previous HGNN method [4] has quasi-linear time complexity (scalability). Besides, the previous method [3] has proposed the HGSampling technique, which helps it scale to a large graph that has 178,663,927 nodes and 2,236,196,802 edges. Please see Table 1 of [4], the OAG dataset is much larger than the Ogbn-mag dataset used in this work.

- Referring to Eq. (5), the proposed method needs to compute the graph structure information propagation for a k-hop meta-path by the matrix multiplication operation between a sequence of adjacency matrices. The time complexity of this operation is quite high, which should be included in the total time complexity.

- In the abstract, the authors claim that their framework achieves the best performance on Ogbn-mag of Open Graph Benchmark. There is a risk of violating the anonymity of double-blind review since the real names appear on the OGB Leaderboard. Besides, I did not see "HGAMLP" on the Leaderboard.


Refs:

[1] [WWW 2019] [HAN] Heterogeneous Graph Attention Network

[2] [NIPS 2019] [GTN] Graph Transformer Networks

[3] [WWW 2020] [HGT] Heterogeneous Graph Transformer

[4] [TKDE 2021] [ie-HGCN] Interpretable and Efficient Heterogeneous Graph Convolutional Network

**Questions:**

Please reply to the weaknesses listed in the previous text box.

**Details Of Ethics Concerns:**

I have no ethics concerns about this work.

---

### Official Review · Reviewer_W1yU · 2023-10-30

**Soundness:** 2 fair
**Presentation:** 3 good
**Contribution:** 2 fair
**Rating:** 3
**Confidence:** 4

**Summary:**

The paper analyzes the limitation of existing scalable HGNNs and proposes a new method called Heterogeneous Graph Attention Multi-Layer prceptron (HGAMLP), which employs a local multi-knowledge extractor, the de-redundancy mechanism and a node-adaptive weight adjustment mechanism to enhance the performance of HGNNs. Experimental results demonstrate the effectiveness of HGAMLP.

**Strengths:**

1. This work focuses on a valuable topic about the scalability of heterogeneous graph learning. Two problems of existing work are presented: fixed knowledge extractor and buried global information.

2. New methods are given based on adequate analysis. The designed de-redundancy mechanism and node-adaptive weight adjustment mechanism are effective.

3. HGAMLP achieves better performance and efficiency using uncomplicated model structure.

**Weaknesses:**

1. The proposed insights are not very convincing. For example, Figure 2 merely illustrates the performance of existing models drops or plateaus as the number of hops increases. It is unintuitive to show that this phenomenon is necessarily due to redundant low-order information.


2. The major concern is the limited novelty. The degree-based normalization method for local multi-knowledge extractor, the mask mechanism for de-redundancy, and the adaptive weight adjustment mechanism for knowledge fusion are simple and commonly-used methods.


3. More larger datasets should be used to demonstrate the scalability and efficiency of HGAMLP.

**Questions:**

1. The main reason that the information about low-order meta-paths is necessarily redundant for high-order graph structure information is not very clear. Is it possible that some of the higher-order information being used requires that lower-order information be considered together?

2. The accuracy of existing scalable HGNNs drops or plateaus as the number of hops increases. Could this be due to something other than low-order redundant information, such as limited information transfer over long distances, error, and noise accumulation, or over-smoothing?


3. Is the number of local knowledge extractors $n$ you used a hyperparameter? Does this number affect the effectiveness of your model?

4. Figure 8 only goes to show that the de-redundancy mechanism is critical to your devised model. However, the performance degradation of existing methods is not necessarily due to this reason.

---

### Official Review · Reviewer_TS4L · 2023-10-31

**Soundness:** 3 good
**Presentation:** 3 good
**Contribution:** 2 fair
**Rating:** 5
**Confidence:** 5

**Summary:**

This paper addresses the issues of redundancy properties of high-order meta-paths and the limitations of fixed knowledge extractors in existing scalable HGNNs. The proposed framework HGAMLP designs a de-redundancy mechanism to extract the pure high-order graph structure information. It also employs a local multi-knowledge extractor and a node-adaptive weight adjustment mechanism to fuse knowledge. Experiments on five graph datasets achieve SOTA performance in both accuracy and training speed.

**Strengths:**

1.	The study of scalable HGNNs is significant and practical. The proposed method is scalable and effective.
2.	The experiments are sufficient to demonstrate the effectiveness of the proposed model.
3.	Most of the paper is easy to follow.

**Weaknesses:**

1.	The motivation is a little weak. The limitation of fixed knowledge extractor and buried global information is not so intuitive and is not closely related to the scalability of HGNNs. The observations and insights are not clear. For example, the stds in attention values are just cases in HGB and can’t represent all attention-based methods.
2.	The method lacks novelty. The de-redundancy mechanism is a little trivial and like a trick. The local multi-knowledge extractor is a simple extension of GNNs to HGNNs and lacks of dedicated design. The node-adaptive weight adjustment lacks of reasonable illustrations and reflects no efficient and scalable properties.
3.	The experiment lacks a SOTA baseline in ogbn-mag leaderboard, i.e., LDMLP [1]. Besides, it only conducts experiments on one large-scale dataset. Since this paper focuses on scalable HGNNs, it’s expected to add more large-scale datasets, e.g., WikiKG [2].

Reference

[1] Li et al. Long-range Dependency based Multi-Layer Perceptron for Heterogeneous Information Networks. 2023

[2] https://ogb.stanford.edu/docs/lsc/

**Questions:**

1.	It’s better to compare the model performance under different aggregators in Figure 3. The stds of attention values depicted in Figure 3 can’t illustrate the mean aggregator is limited. In Figure 4(a), an attention mechanism is just the solution for different nodes requiring different meta-paths to achieve high accuracy. Besides, it’s confused that different colors denote different meta-paths in Figure 4(b). In my view, the color in each cell should denote the redundant degree as illustrated by the author.
2.	Why the output of adaptive global knowledge fusion module is called X_local? What the global and local means in concrete? It’s better to illustrate this more clearly. In Table 2, the trick column is suggested to be removed and adding illustrations in caption.